# Graphical Generative Adversarial Networks

**Chongxuan Li**[*]
licx14@mails.tsinghua.edu.cn

**Max Welling**[†]
M.Welling@uva.nl

**Jun Zhu**[*]
dcszj@mail.tsinghua.edu.cn

**Bo Zhang**[*]
dcszb@mail.tsinghua.edu.cn

## Abstract

We propose Graphical Generative Adversarial Networks (Graphical-GAN) to model structured data. Graphical-GAN conjoins the power of Bayesian networks on compactly representing the dependency structures among random variables and that of generative adversarial networks on learning expressive dependency functions. We introduce a structured recognition model to infer the posterior distribution of latent variables given observations. We generalize the Expectation Propagation (EP) algorithm to learn the generative model and recognition model jointly. Finally, we present two important instances of Graphical-GAN, i.e. Gaussian Mixture GAN (GMGAN) and State Space GAN (SSGAN), which can successfully learn the discrete and temporal structures on visual datasets, respectively.

## 1 Introduction

Deep implicit models [29] have shown promise on synthesizing realistic images [10, 33, 2] and inferring latent variables [26, 11]. However, these approaches do not explicitly model the underlying structures of the data, which are common in practice (e.g., temporal structures in videos). Probabilistic graphical models [18] provide principle ways to incorporate the prior knowledge about the data structures but these models often lack the capability to deal with the complex data like images.

To conjoin the benefits of both worlds, we propose a flexible generative modelling framework called *Graphical Generative Adversarial Networks* (Graphical-GAN). On one hand, Graphical-GAN employs Bayesian networks [18] to represent the structures among variables. On the other hand, Graphical-GAN uses deep implicit likelihood functions [10] to model complex data.

Graphical-GAN is sufficiently flexible to model structured data but the inference and learning are challenging due to the presence of deep implicit likelihoods and complex structures. We build a structured recognition model [17] to approximate the true posterior distribution. We study two families of the recognition models, i.e. the *mean field posteriors* [14] and the *inverse factorizations* [39]. We generalize the *Expectation Propagation* (EP) [27] algorithm to learn the generative model and recognition model jointly. Motivated by EP, we minimize a local divergence between the generative model and recognition model for each individual local factor defined by the generative model. The local divergences are estimated via the adversarial technique [10] to deal with the implicit likelihoods.

Given a specific scenario, the generative model is determined a priori by context or domain knowledge and the proposed inference and learning algorithms are applicable to arbitrary Graphical-GAN. As instances, we present Gaussian Mixture GAN (GMGAN) and State Space GAN (SSGAN) to learn the discrete and temporal structures on visual datasets, respectively. Empirically, these models can

---

[*]Department of Computer Science & Technology, Institute for Artificial Intelligence, BNRist Center, THBI Lab, State Key Lab for Intell. Tech. & Sys., Tsinghua University. Correspondence to: J. Zhu.

[†]University of Amsterdam, and the Canadian Institute for Advanced Research (CIFAR).

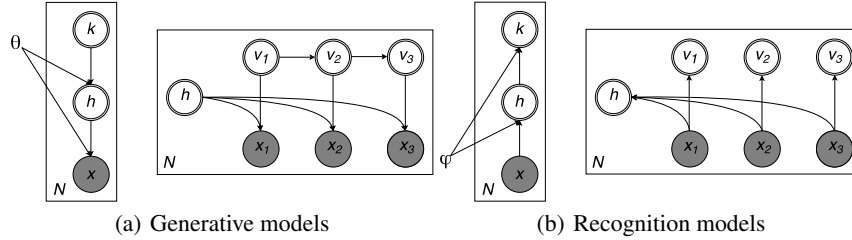

(a) Generative models        (b) Recognition models

Figure 1: (a) Generative models of GMGAN (left panel) and SSGAN (right panel). (b) Recognition models of GMGAN and SSGAN. The grey and white units denote the observed and latent variables, respectively. The arrows denote dependencies between variables. $\theta$ and $\phi$ denote the parameters in the generative model and recognition model, respectively. We omit $\theta$ and $\phi$ in SSGAN for simplicity.

infer the latent structures and generate structured samples. Further, Graphical-GAN outperforms the baseline models on inference, generation and reconstruction tasks consistently and substantially.

Overall, our contributions are: (1) we propose Graphical-GAN, a general generative modelling framework for structured data; (2) we present two instances of Graphical-GAN to learn the discrete and temporal structures, respectively; and (3) we empirically evaluate Graphical-GAN on generative modelling of structured data and achieve good qualitative and quantitative results.

## 2 General Framework

In this section, we present the model definition, inference method and learning algorithm.

### 2.1 Model Definition

Let $X$ and $Z$ denote the observable variables and latent variables, respectively. We assume that we have $N$ i.i.d. samples from a generative model with the joint distribution $p_{\mathcal{G}}(X, Z) = p_{\mathcal{G}}(Z)p_{\mathcal{G}}(X|Z)$, where $\mathcal{G}$ is the associated directed acyclic graph (DAG). According to the local structures of $\mathcal{G}$, the distribution of a single data point can be further factorized as follows:

$$p_{\mathcal{G}}(X, Z) = \prod_{i=1}^{|Z|} p(z_i|\mathrm{pa}_{\mathcal{G}}(z_i)) \prod_{j=1}^{|X|} p(x_j|\mathrm{pa}_{\mathcal{G}}(x_j)), \qquad (1)$$

where $\mathrm{pa}_{\mathcal{G}}(x)$ denotes the parents of $x$ in the associated graph $\mathcal{G}$. Note that only latent variables can be the parent of a latent variable and see Fig 1 (a) for an illustration. Following the factorization in Eqn. (1), we can sample from the generative model efficiently via *ancestral sampling*.

Given the dependency structures, the dependency functions among the variables can be parameterized as deep neural networks to fit complicated data. As for the likelihood functions, we consider *implicit probabilistic models* [29] instead of *prescribed probabilistic models*. Prescribed models [17] define the likelihood functions for $X$ with an explicit specification. In contrast, implicit models [10] deterministically transform $Z$ to $X$ and the likelihood can be intractable. We focus on implicit models because they have been proven effective on image generation [33, 2] and the learning algorithms for implicit models can be easily extended to prescribed models. We also directly compare with existing structured prescribed models [7] in Sec. 5.1. Following the well established literature, we refer to our model as Graphical Generative Adversarial Networks (Graphical-GAN).

The inference and learning of Graphical-GAN are nontrivial. On one hand, Graphical-GAN employs deep implicit likelihood functions, which makes the inference of the latent variables intractable and the likelihood-based learning method infeasible. On the other hand, Graphical-GAN involves complex structures, which requires the inference and learning algorithm to exploit the structural information explicitly. To address the problems, we propose structured recognition models and a sample-based massage passing algorithm, as detailed in Sec. 2.2 and Sec. 2.3, respectively.

## 2.2 Inference Method

We leverage recent advances on *amortized inference* of deep generative models [17, 9, 8, 40] to infer the latent variables given the data. Basically, these approaches introduce a recognition model, which is a family of distributions of a simple form, to approximate the true posterior. The recognition model is shared by all data points and often parameterized as a deep neural network.

The problem is more complicated in our case because we need to further consider the graphical structure during the inference procedure. Naturally, we introduce a structured recognition model with an associated graph $\mathcal{H}$ as the approximate posterior, whose distribution is formally given by:

$$q_{\mathcal{H}}(Z|X) = \prod_{i=1}^{|Z|} q(z_i|\mathrm{pa}_{\mathcal{H}}(z_i)). \tag{2}$$

Given data points from the true data distribution $q(X)$, we can obtain samples following the joint distribution $q_{\mathcal{H}}(X, Z) = q(X)q_{\mathcal{H}}(Z|X)$ efficiently via ancestral sampling. Considering different dependencies among the variables, or equivalently $\mathcal{H}$s, we study two types of recognition models: the mean-field posteriors [14] and the inverse factorizations [39].

The mean-field assumption has been widely adopted to variational inference methods [14] because of its simplicity. In such methods, all of the dependency structures among the latent variables are ignored and the approximate posterior could be factorized as follows:

$$q_{\mathcal{H}}(Z|X) = \prod_{i=1}^{|Z|} q(z_i|X), \tag{3}$$

where the associated graph $\mathcal{H}$ has fully factorized structures.

The inverse factorizations [39] approach views the original graphical model as a *forward factorization* and samples the latent variables given the observations efficiently by inverting $\mathcal{G}$ step by step. Formally, the *inverse factorization* is defined as follows:

$$q_{\mathcal{H}}(Z|X) = \prod_{i=1}^{|Z|} q(z_i|\partial_{\mathcal{G}}(z_i) \cap z_{>i}), \tag{4}$$

where $\partial_{\mathcal{G}}(z_i)$ denotes the *Markov blanket* of $z_i$ on $\mathcal{G}$ and $z_{>i}$ denotes all $z$ after $z_i$ in a certain order, which is defined from leaves to root according to the structure of $\mathcal{G}$. See the formal algorithm to build $\mathcal{H}$ based on $\mathcal{G}$ in Appendix A.

Given the structure of the approximate posterior, we also parameterize the dependency functions as neural networks of similar sizes to those in the generative models. Both posterior families are generally applicable for arbitrary Graphical-GANs and we use them in two different instances, respectively. See Fig. 1 (b) for an illustration.

## 2.3 Learning Algorithm

Let $\theta$ and $\phi$ denote the parameters in the generative model, $p$, and the recognition model $q$, respectively. Our goal is to learn $\theta$ and $\phi$ jointly via divergence minimization, which is formulated as:

$$\min_{\theta, \phi} \mathcal{D}(q(X, Z) || p(X, Z)), \tag{5}$$

where we omit the subscripts of the associated graphs in $p$ and $q$ for simplicity. We restrict $\mathcal{D}$ in the $f$-divergence family [5], that is $\mathcal{D}(q(X, Z) || p(X, Z)) = \int p(X, Z) f(\frac{q(X,Z)}{p(X,Z)}) dX dZ$, where $f$ is a convex function of the likelihood ratio. The Kullback-Leibler (KL) divergence and the Jensen-Shannon (JS) divergence are included.

Note that we cannot optimize Eqn. (5) directly because the likelihood ratio is unknown given implicit $p(X, Z)$. To this end, ALI [8, 9] introduces a parametric discriminator to estimate the divergence via discriminating the samples from the models. We can directly apply ALI to Graphical-GAN by treating all variables as a whole and we refer it as the global baseline (See Appendix B for the formal algorithm). The global baseline uses a single discriminator that takes all variables as input. It may be

sub-optimal in practice because the capability of a single discriminator is insufficient to distinguish complex data, which makes the estimate of the divergence not reliable. Intuitively, the problem will be easier if we exploit the data structures explicitly when discriminating the samples. The intuition motivates us to propose a local algorithm like Expectation Propagation (EP) [27], which is known as a deterministic approximation algorithm with analytic and computational advantages over other approximations, including Variational Inference [21].

Following EP, we start from the factorization of $p(X, Z)$ in terms of a set of factors $F_{\mathcal{G}}$:

$$p(X, Z) \propto \prod_{A \in F_{\mathcal{G}}} p(A). \tag{6}$$

Generally, we can choose any reasonable $F_{\mathcal{G}}^{3}$ but here we specify that $F_{\mathcal{G}}$ consists of families $(x, \mathrm{pa}_{\mathcal{G}}(x))$ and $(z, \mathrm{pa}_{\mathcal{G}}(z))$ for all $x$ and $z$ in the model. We assume that the recognition model can also be factorized in the same way. Namely, we have

$$q(X, Z) \propto \prod_{A \in F_{\mathcal{G}}} q(A). \tag{7}$$

Instead of minimizing Eqn. (5), EP iteratively minimizes a *local divergence* in terms of each factor individually. Formally, for factor $A$, we're interested in the following divergence [27, 28]:

$$\mathcal{D}(q(A)\overline{q(A)}||p(A)\overline{p(A)}), \tag{8}$$

where $\overline{p(A)}$ denotes the marginal distribution over the complementary $\bar{A}$ of $A$. EP [27] further assmues that $\overline{q(A)} \approx \overline{p(A)}$ to make the expression tractable. Though the approximation cannot be justified theoretically, empirical results [28] suggest that the gap is small if the approximate posterior is a good fit to the true one. Given the approximation, for each factor $A$, the objective function changes to:

$$\mathcal{D}(q(A)\overline{q(A)}||p(A)\overline{q(A)}). \tag{9}$$

Here we make the same assumption because $\overline{q(A)}$ will be cancelled in the likelihood ratio if $\mathcal{D}$ belongs to $f$-divergence and we can ignore other factors when checking factor $A$, which reduces the complexity of the problem. For instance, we can approximate the JS divergence for factor $A$ as:

$$\mathcal{D}_{JS}(q(X, Z)||p(X, Z)) \approx \mathbb{E}_q[\log \frac{q(A)}{m(A)}] + \mathbb{E}_p[\log \frac{p(A)}{m(A)}], \tag{10}$$

where $m(A) = \frac{1}{2}(p(A) + q(A))$. See Appendix C for the derivation. As we are doing amortized inference, we further average the divergences over all local factors as:

$$\frac{1}{|F_{\mathcal{G}}|} \sum_{A \in F_{\mathcal{G}}} \left[ \mathbb{E}_q[\log \frac{q(A)}{m(A)}] + \mathbb{E}_p[\log \frac{p(A)}{m(A)}] \right] = \frac{1}{|F_{\mathcal{G}}|} \left[ \mathbb{E}_q[\sum_{A \in F_{\mathcal{G}}} \log \frac{q(A)}{m(A)}] + \mathbb{E}_p[\sum_{A \in F_{\mathcal{G}}} \log \frac{p(A)}{m(A)}] \right]. \tag{11}$$

The equality holds due to the linearity of the expectation. The expression in Eqn. (11) provides an efficient solution where we can obtain samples over the entire variable space once and repeatedly project the samples into each factor. Finally, we can estimate the local divergences using individual discriminators and the entire objective function is as follows:

$$\max_{\psi} \frac{1}{|F_{\mathcal{G}}|} \mathbb{E}_q[\sum_{A \in F_{\mathcal{G}}} \log(D_A(A))] + \frac{1}{|F_{\mathcal{G}}|} \mathbb{E}_p[\sum_{A \in F_{\mathcal{G}}} \log(1 - D_A(A))], \tag{12}$$

where $D_A$ is the discriminator for the factor $A$ and $\psi$ denotes the parameters in all discriminators.

Though we assume that $q(X, Z)$ shares the same factorization with $p(X, Z)$ as in Eqn. (7) when deriving the objective function, the result in Eqn. (12) does not specify the form of $q(X, Z)$. This is because we do not need to compute $q(A)$ explicitly and instead we directly estimate the likelihood ratio based on samples. This makes it possible for Graphical-GAN to use an arbitrary $q(X, Z)$, including the two recognition models presented in Sec. 2.2, as long as we can sample from it quickly.

Given the divergence estimate, we perform the stochastic gradient decent to update the parameters. We use the reparameterization trick [17] and the Gumbel-Softmax trick [12] to estimate the gradients with continuous and discrete random variables, respectively. We summarize the procedure in Algorithm 1.

## 3 Two Instances

We consider two common and typical scenarios involving structured data in practice. In the first one, the dataset consists of images with discrete attributes or classes but the groundtruth for an individual sample is unknown. In the second one, the dataset consists of sequences of images with temporal dependency within each sequence. We present two important instances of Graphical-GAN, i.e. Gaussian Mixture GAN (GM-GAN) and State Space GAN (SSGAN), to deal with these two scenarios, respectively. These instances show the abilities of our general framework to deal with discrete latent variables and complex structures, respectively.

---
**Algorithm 1** Local algorithm for Graphical-GAN
---
**repeat**
 • Get a minibatch of samples from $p(X, Z)$
 • Get a minibatch of samples from $q(X, Z)$
 • Approximate the divergence $\mathcal{D}(q(X, Z)||p(X, Z))$ using Eqn. (12) and the current value of $\psi$
 • Update $\psi$ to maximize the divergence
 • Get a minibatch of samples from $p(X, Z)$
 • Get a minibatch of samples from $q(X, Z)$
 • Approximate the divergence $\mathcal{D}(q(X, Z)||p(X, Z))$ using Eqn. (12) and the current value of $\psi$
 • Update $\theta$ and $\phi$ to minimize the divergence
**until** Convergence or reaching certain threshold

---

**GMGAN** We assume that the data consists of $K$ mixtures and hence uses a mixture of Gaussian prior. Formally, the generative process of GMGAN is:

$$k \sim Cat(\pi), h|k \sim \mathcal{N}(\mu_k, \Sigma_k), x|h = G(h),$$

where $Z = (k, h)$, and $\pi$ and $G$ are the coefficient vector and the generator, respectively. We assume that $\pi$ and $\Sigma_k$s are fixed as the uniform prior and identity matrices, respectively. Namely, we only have a few extra trainable parameters, i.e. the means for the mixtures $\mu_k$s.

We use the inverse factorization as the recognition model because it preserves the dependency relationships in the model. The resulting approximate posterior is a simple inverse chain as follows:

$$h|x = E(x), q(k|h) = \frac{\pi_k \mathcal{N}(h|\mu_k, \Sigma_k)}{\sum_{k'} \pi_{k'} \mathcal{N}(h|\mu_{k'}, \Sigma_{k'})},$$

where $E$ is the extractor that maps data points to the latent variables.

In the global baseline, a single network is used to discriminate the $(x, h, k)$ tuples. In our local algorithm, two separate networks are introduced to discriminate the $(x, h)$ and $(h, k)$ pairs, respectively.

**SSGAN** We assume that there are two types of latent variables. One is invariant across time, denoted as $h$ and the other varies across time, denoted as $v_t$ for time stamp $t = 1, ..., T$. Further, SSGAN assumes that $v_t$s form a Markov Chain. Formally, the generative process of SSGAN is:

$$v_1 \sim \mathcal{N}(0, I), h \sim \mathcal{N}(0, I), \qquad \epsilon_t \sim \mathcal{N}(0, I), \forall t = 1, 2, ..., T - 1,$$
$$v_{t+1}|v_t = O(v_t, \epsilon_t), \forall t = 1, 2, ..., T - 1, \qquad x_t|h, v_t = G(h, v_t), \forall t = 1, 2, ..., T,$$

where $Z = (h, v_1, ..., v_T)$, and $O$ and $G$ are the transition operator and the generator, respectively. They are shared across time under the stationary and output independent assumptions, respectively.

For simplicity, we use the mean-field recognition model as the approximate posterior:

$$h|x_1, x_2..., x_T = E_1(x_1, x_2..., x_T), \qquad v_t|x_1, x_2..., x_T = E_2(x_t), \forall t = 1, 2, ..., T,$$

where $E_1$ and $E_2$ are the extractors that map the data points to $h$ and $v$ respectively. $E_2$ is also shared across time.

In the global baseline, a single network is used to discriminate the $(x_1, ..., x_T, v_1, ..., v_T, h)$ samples. In our local algorithm, two separate networks are introduced to discriminate the $(v_t, v_{t+1})$ pairs and $(x_t, v_t, h)$ tuples, respectively. Both networks are shared across time, as well.

## 4 Related Work

**General framework** The work of [13, 16, 22] are the closest papers on the structured deep generative models. Johnson et al. [13] introduce structured Bayesian priors to Variational Auto-Encoders

(VAE) [17] and propose efficient inference algorithms with conjugated exponential family structure. Lin et al. [22] consider a similar model as in [13] and derive an amortized variational message passing algorithm to simplify and generalize [13]. Compared to [13, 22], Graphical-GAN is more flexible on the model definition and learning methods, and hence can deal with natural data.

Adversarial Massage Passing (AMP) [16] also considers structured implicit models but there exist several key differences to make our work unique. Theoretically, Graphical-GAN and AMP optimize different local divergences. As presented in Sec. 2.3, we follow the recipe of EP precisely to optimize $\mathcal{D}(q(A)\overline{q(A)}||p(A)\overline{q(A)})$ and naturally derive our algorithm that involves only the factors defined by $p(X, Z)$, e.g. $A = (z_i, \mathrm{pa}_{\mathcal{G}}(z_i))$. On the other hand, AMP optimizes another local divergence $\mathcal{D}(q(A')||p(A))$, where $A'$ is a factor defined by $q(X, Z)$, e.g. $A' = (z_i, \mathrm{pa}_{\mathcal{H}}(z_i))$. In general, $A'$ can be different from $A$ because the DAGs $\mathcal{G}$ and $\mathcal{H}$ have different structures. Further, the theoretical difference really matters in practice. In AMP, the two factors involved in the local divergence are defined over different domains and hence may have different dimensionalities generally. Therefore, it remains unclear how to implement AMP [4] because a discriminator cannot take two types of inputs with different dimensionalities. In fact, no empirical evidence is reported in AMP [16]. In contrast, Graphical-GAN is easy to implement by considering only the factors defined by $p(X, Z)$ and achieves excellent empirical results (See Sec. 5).

There is much work on the learning of implicit models. $f$-GAN [31] and WGAN [2] generalize the original GAN using the $f$-divergence and Wasserstein distance, respectively. The work of [40] minimizes a penalized form of the Wasserstein distance in the optimal transport point of view and naturally considers both the generative modelling and inference together. The Wasserstein distance can also be used in Graphical-GAN to generalize our current algorithms and we leave it for the future work. The recent work of [34] and [41] perform Bayesian learning for GANs. In comparison, Graphical-GAN focuses on learning a probabilistic graphical model with latent variables instead of posterior inference on global parameters.

**Instances** Several methods have learned the discrete structures in an unsupervised manner. Makhzani et al. [24] extend an autoencoder to a generative model by matching the *aggregated posterior* to a prior distribution and shows the ability to cluster handwritten digits. [4] introduce some interpretable codes independently from the other latent variables and regularize the original GAN loss with the mutual information between the codes and the data. In contrast, GMGAN explicitly builds a hierarchical model with top-level discrete codes and no regularization is required. The most direct competitor [7] extends VAE [17] with a mixture of Gaussian prior and is compared with GMGAN in Sec. 5.1.

There exist extensive prior methods on synthesizing videos but most of them condition on input frames [38, 32, 25, 15, 46, 43, 6, 42]. Three of these methods [44, 35, 42] can generate videos without input frames. In [44, 35], all latent variables are generated jointly and without structure. In contrast, SSGAN explicitly disentangles the invariant latent variables from the variant ones and builds a Markov chain on the variant ones, which makes it possible to do motion analogy and generalize to longer sequences. MoCoGAN [42] also exploits the temporal dependency of the latent variables via a recurrent neural network but it requires heuristic regularization terms and focuses on generation. In comparison, SSGAN is an instance of the Graphical-GAN framework, which provides theoretical insights and a recognition model for inference.

Compared with all instances, Graphical-GAN does not focus on a specific structure, but provides a general way to deal with arbitrary structures that can be encoded as Bayesian networks.

## 5 Experiments

We implement our model using the TensorFlow [1] library.[5] In all experiments, we optimize the JS-divergence. We use the widely adopted DCGAN architecture [33] in all experiments to fairly compare Graphical-GAN with existing methods. We evaluate GMGAN on the MNIST [20], SVHN [30], CIFAR10 [19] and CelebA [23] datasets. We evaluate SSGAN on the Moving MNIST [38] and 3D chairs [3] datasets. See Appendix D for further details of the model and datasets.

In our experiments, we are going to show that

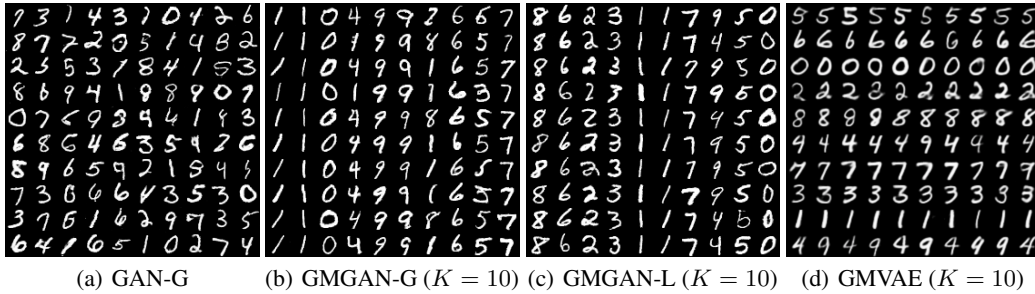

| (a) GAN-G | (b) GMGAN-G ($K = 10$) | (c) GMGAN-L ($K = 10$) | (d) GMVAE ($K = 10$) |

Figure 2: Samples on the MNIST dataset. The results of (a) are comparable to those reported in [8]. The mixture $k$ is fixed in each column of (b) and (c). $k$ is fixed in each row of (d), which is from [7].

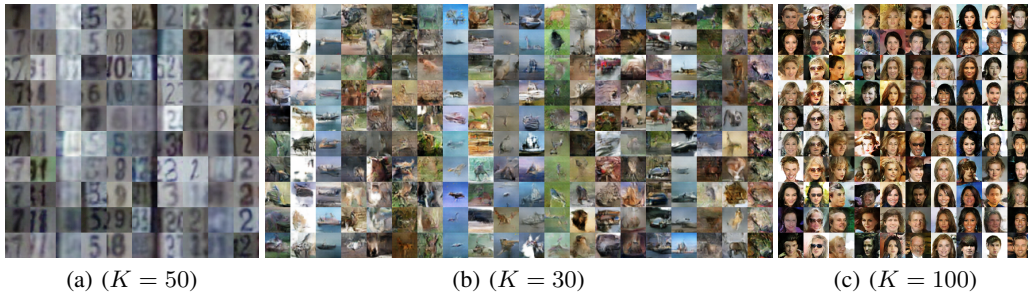

| (a) ($K = 50$) | (b) ($K = 30$) | (c) ($K = 100$) |

Figure 4: Part of samples of GMGAN-L on SVHN (a) CIFAR10 (b) and CelebA (c) datasets. The mixture $k$ is fixed in each column. See the complete results in Appendix E.

- Qualitatively, Graphical-GAN can infer the latent structures and generate structured samples without any regularization, which is required by existing models [4, 43, 6, 42];
- Quantitatively, Graphical-GAN can outperform all baseline methods [7–9] in terms of inference accuracy, sample quality and reconstruction error consistently and substantially.

## 5.1 GMGAN Learns Discrete Structures

We focus on the unsupervised learning setting in GMGAN. Our assumption is that there exist discrete structures, e.g. classes and attributes, in the data but the ground truth is unknown. We compare Graphical-GAN with three existing methods, i.e. ALI [8, 9], GMVAE [38] and the global baseline. For simplicity, we denote the global baseline and our local algorithm as *GMGAN-G* and *GMGAN-L*, respectively. Following this, we also denote ALI as *GAN-G*.

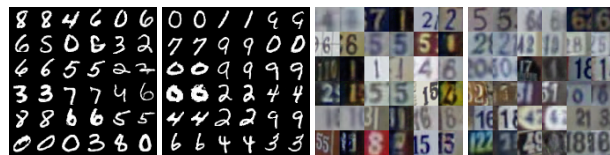

| (a) GAN-G | (b) GMGAN-L | (c) GAN-G | (d) GMGAN-L |

Figure 3: Reconstruction on the MNIST and SVHN datasets. Each odd column shows the test inputs and the next even column shows the corresponding reconstruction. (a) and (c) are comparable to those reported in [8, 9].

We first compare the samples of all models on the MNIST dataset in Fig. 2. As for sample quality, GMGAN-L has less meaningless samples compared with GAN-G (i.e. ALI), and has sharper samples than those of the GMVAE. Besides, as for clustering performance, GMGAN-L is superior to GMGAN-G and GMVAE with less ambiguous clusters. We then demonstrate the ability of GMGAN-L to deal with more challenging datasets. The samples on the SVHN, CIFAR10 and CelebA datasets are shown in Fig. 4. Given a fixed mixture $k$, GMGAN-L can generate samples with similar semantics and visual factors, including the object classes, backgrounds and attributes like

Table 1: The clustering accuracy (ACC) [37], inception score (IS) [36] and mean square error (MSE) results for inference, generation and reconstruction tasks, respectively. The results of our implementation are averaged over 10 (ACC) or 5 (IS and MSE) runs with different random seeds.

| Algorithm | ACC on MNIST | IS on CIFAR10 | MSE on MNIST |
|---|---|---|---|
| *GMVAE* | 92.77 ($\pm$1.60) [7] | - | - |
| *CatGAN* | 90.30 [37] | - | - |
| *GAN-G* | - | 5.34 ($\pm$0.05) [45] | - |
| *GMM* (our implementation) | 68.33($\pm$0.21) | - | - |
| *GAN-G+GMM* (our implementation) | 70.27($\pm$0.50) | 5.26 ($\pm$0.05) | 0.071 ($\pm$0.001) |
| *GMGAN-G* (our implementation) | 91.62 ($\pm$1.91) | 5.41 ($\pm$0.08) | 0.056 ($\pm$0.001) |
| *GMGAN-L* (ours) | **93.03** ($\pm$1.65) | **5.94** ($\pm$0.06) | **0.044** ($\pm$0.001) |

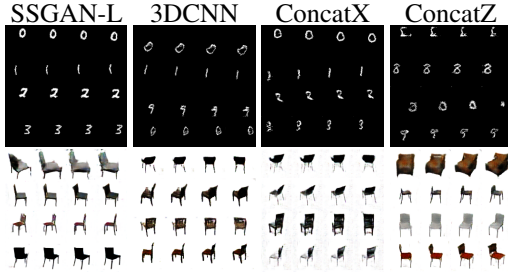

Figure 5: Samples on the Moving MNIST and 3D chairs datasets when $T = 4$. Each row in a subfigure represents a video sample.

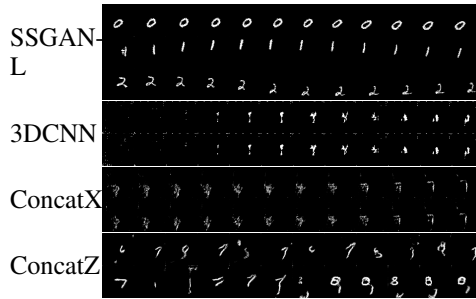

Figure 6: Samples (first 12 frames) on the Moving MNIST dataset when $T = 16$.

"wearing glasses". We also show the samples of GMGAN-L by varying $K$ and linearly interpolating the latent variables in Appendix E.

We further present the reconstruction results in Fig. 3. GMGAN-L outperforms GAN-G significantly in terms of preserving the same semantics and similar visual appearance. Intuitively, this is because the Gaussian mixture prior helps the model learn a more spread latent space with less ambiguous areas shared by samples in different classes. We empirically verify the intuition by visualizing the latent space via the t-SNE algorithm in Appendix E.

Finally, we compare the models on inference, generation and reconstruction tasks in terms of three widely adopted metrics in Tab. 1. As for the clustering accuracy, after clustering the test samples, we first find the sample that is nearest to the centroid of each cluster and use the label of that sample as the prediction of the testing samples in the same cluster following [37]. GAN-G cannot cluster the data directly, and hence we train a Gaussian mixture model (GMM) on the latent space of GAN-G and the two-stage baseline is denoted as *GAN-G + GMM*. We also train a GMM on the raw data as the simplest baseline. For the GMM implementation, we use the sklearn package and the settings are same as our Gaussian mixture prior. AAE [24] achieves higher clustering accuracy while it is less comparable to our method. Nevertheless, GMGAN-L outperforms all baselines consistently, which agrees with the qualitative results. We also provide the clustering results on the CIFAR10 dataset in Appendix E.

## 5.2 SSGAN Learns Temporal Structures

We denote the SSGAN model trained with the local algorithm as *SSGAN-L*. We construct three types of baseline models, which are trained with the global baseline algorithm but use discriminators with different architectures. The *ConcatX* baseline concatenates all input frames together and processes the input as a whole image with a 2D CNN. The *ConcatZ* baseline processes the input frames independently using a 2D CNN and concatenates the features as the input for fully connected layers to obtain the latent variables. The *3DCNN* baseline uses a 3D CNN to process the whole input directly. In particular, the 3DCNN baseline is similar to existing generative models [44, 35]. The

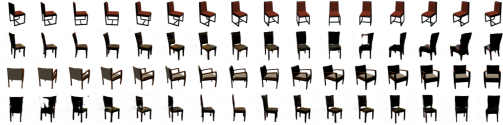 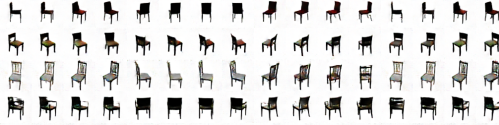

Figure 7: Motion analogy results. Each odd row shows an input and the next even row shows the sample.

Figure 8: 16 of 200 frames generated by SSGAN-L. The frame indices are 47-50, 97-100, 147-150 and 197-200 from left to right in each row.

key difference is that we omit the two stream architecture proposed in [44] and the singular value clipping proposed in [35] for fair comparison as our contribution is orthogonal to these techniques. Also note that our problem is more challenging than those in existing methods [44, 35] because the discriminator in Graphical-GAN needs to discriminate the latent variables besides the video frames.

All models can generate reasonable samples of length 4 on both Moving MNIST and 3D chairs datasets, as shown in Fig. 5. However, if the structure of the data gets complicated, i.e. $T = 16$ on Moving MNIST and $T = 31$ on 3D chairs, all baseline models fail while SSGAN-L can still successfully generate meaningful videos, as shown in Fig. 6 and Appendix F, respectively. Intuitively, this is because a single discriminator cannot provide reliable divergence estimate with limited capability in practise. See the reconstruction results of SSGAN-L in Appendix F.

Compared with existing GAN models [44, 35, 42] on videos, SSGAN-L can learn interpretable features thanks to the factorial structure in each frame. We present the motion analogy results on the 3D chairs dataset in Fig. 7. We extract the variant features $v$, i.e. the motion, from the input testing video and provide a fixed invariant feature $h$, i.e. the content, to generate samples. The samples can track the motion of the corresponding input and share the same content at the same time. Existing methods [43, 6] on learning interpretable features rely on regularization terms to ensure the disentanglement while SSGAN uses a purely adversarial loss.

Finally, we show that though trained on videos of length 31, SSGAN can generate much longer sequences of 200 frames in Fig. 8 thanks to the Markov structure, which again demonstrates the advantages of SSGAN over existing generative models [44, 35, 42].

## 6 Conclusion

This paper introduces a flexible generative modelling framework called Graphical Generative Adversarial Networks (Graphical-GAN). Graphical-GAN provides a general solution to utilize the underlying structural information of the data. Empirical results of two instances show the promise of Graphical-GAN on learning interpretable representations and generating structured samples. Possible extensions to Graphical-GAN include: generalized learning and inference algorithms, instances with more complicated structures (e.g., trees) and semi-supervised learning for structured data.

**Acknowledgments**

The work was supported by the National Key Research and Development Program of China (No. 2017YFA0700900), the National NSF of China (Nos. 61620106010, 61621136008, 61332007), the MIIT Grant of Int. Man. Comp. Stan (No. 2016ZXFB00001), the Youth Top-notch Talent Support Program, Tsinghua Tiangong Institute for Intelligent Computing, the NVIDIA NVAIL Program and a Project from Siemens. This work was done when C. Li visited the university of Amsterdam. During this period, he was supported by China Scholarship Council.

## Footnotes

[3]For instance, we can specify that $F_{\mathcal{G}}$ has only one factor that involves all variables, which reduces to ALI.

[4]Despite our best efforts to contact the authors we did not receive an answer of the issue.

[5]Our source code is available at https://github.com/zhenxuan00/graphical-gan.

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
