[Supplementary Material · appendix.pdf]

---

**Algorithm 1** Inverse factorization

---
**Input** the associated graph $\mathcal{G}$ of $p_{\mathcal{G}}(X, Z)$
**Output** the inverse factorization $\mathcal{H}$ of $q_{\mathcal{H}}(Z|X)$
• Order the latent variables $Z$ from leaves to roots according to $\mathcal{G}$
• Initialize $\mathcal{H}$ as all observable variables $X$ without any edge
**for** $z_i \in Z$ **do**
    • Add $z_i$ to $\mathcal{H}$ and set $\mathrm{pa}_{\mathcal{H}}(z_i) = \partial_{\mathcal{G}}(z_i) \cap \mathcal{H}$
**end for**

---

---

**Algorithm 2** The global baseline for Graphical-GAN

---
**repeat**
    • Get a minibatch of samples from $p(X, Z)$
    • Get a minibatch of samples from $q(X, Z)$
    • Estimate the divergence $\mathcal{D}(q(X, Z)||p(X, Z))$ using Eqn. (1) and the current value of $\psi$
    • Update $\psi$ to maximize the divergence
    • Get a minibatch of samples from $p(X, Z)$
    • Get a minibatch of samples from $q(X, Z)$
    • Estimate the divergence $\mathcal{D}(q(X, Z)||p(X, Z))$ using Eqn. (1) and the current value of $\psi$
    • Update $\theta$ and $\phi$ to minimize the divergence
**until** Convergence or reaching certain threshold

---

## A   Algorithm for Inverse Factorizations

We present the formal procedure of building the inverse factorizations in Alg. 1.

## B   The Global Baseline

We can directly adopt ALI to learn Graphical-GAN. Formally, the estimate of the divergence is given by:

$$\max_{\psi} \mathbb{E}_q[\log(D(X, Z))] + \mathbb{E}_p[\log(1 - D(X, Z))], \tag{1}$$

where $D$ is the discriminator introduced for divergence estimation and $\psi$ denotes the parameters in $D$. If $D$ is Bayes optimal, then the estimate actually equals to $2\mathcal{D}_{JS}(q(X, Z)||p(X, Z)) - \log 4$, which is equivalent to $\mathcal{D}_{JS}(q(X, Z)||p(X, Z))$. We present the formal procedure of building the inverse factorizations in Alg. 2.

## C   The Local Approximation of the JS Divergence

We now derive the local approximation of the JS divergence.

$$
\begin{aligned}
&\mathcal{D}_{JS}(q(X, Z)||p(X, Z)) \\
\approx &\mathcal{D}_{JS}(q(A)\overline{q(A)}||p(A)\overline{p(A)}) \\
\approx &\mathcal{D}_{JS}(q(A)\overline{q(A)}||p(A)\overline{q(A)}) \\
= &\int q(A)\overline{q(A)} \log \frac{q(A)\overline{q(A)}}{\frac{p(A)\overline{q(A)}+q(A)\overline{q(A)}}{2}} dXdZ + \int p(A)\overline{q(A)} \log \frac{p(A)\overline{q(A)}}{\frac{p(A)\overline{q(A)}+q(A)\overline{q(A)}}{2}} dXdZ \\
= &\int q(A)\overline{q(A)} \log \frac{q(A)}{m(A)} dXdZ + \int p(A)\overline{q(A)} \log \frac{p(A)}{m(A)} dXdZ \\
\approx &\int q(A)\overline{q(A)} \log \frac{q(A)}{m(A)} dXdZ + \int p(A)\overline{p(A)} \log \frac{p(A)}{m(A)} dXdZ \\
\approx &\mathbb{E}_q \log \frac{q(A)}{m(A)} + \mathbb{E}_p \log \frac{p(A)}{m(A)},
\end{aligned}
$$

12 where $m(A) = \frac{1}{2}(p(A) + q(A))$. As for the approximations, we adopt two commonly used assump-
13 tions: (1) $\overline{q(A)} \approx \overline{p(A)}$, and (2) $p(X, Z) \approx p(A)\overline{p(A)}$ and $q(X, Z) \approx q(A)\overline{q(A)}$.

## D  Experimental Details

15 We evaluate GMGAN on the MNIST, SVHN , CIFAR10 and CelebA datasets. The MNIST dataset
16 consists of handwritten digits of size 28×28 and there are 50,000 training samples, 10,000 validation
17 samples and 10,000 testing samples. The SVHN dataset consists of digit sequences of size 32×32,
18 and there are 73,257 training samples and 26,032 testing samples. The CIFAR10 dataset consists of
19 natural images of size 32×32 and there are 50,000 training samples and 10,000 testing samples. The
20 CelebA dataset consists of 202,599 faces and we randomly sample 5,000 samples for testing. Further,
21 the faces are center cropped and downsampled to size 64×64.

22 We evaluate SSGAN on the Moving MNIST and 3D chairs datasets. In the Moving MNIST dataset,
23 each clip contains a handwritten digit which bounces inside a 64×64 patch. The velocity of the
24 digit is randomly sampled and fixed within a clip. We generate two datasets of length 4 and length
25 16, respectively. To improve the sample quality, all models condition on the label information as in
26 DCGAN. The 3D chairs dataset consists of 2,786 sequences of rendered chairs and each sequence is
27 of length 31. We randomly sample sub-sequences if necessary. The clips in the 3D chairs dataset are
28 center cropped and downsampled to size 64×64. No supervision is used on the 3D chairs dataset.

Table 1: The generator and extractor on **SVHN**

| Generator $G$ | Extractor $E$ |
|---|---|
| Input latent $h$ | Input image $x$ |
| MLP 4096 units ReLU | 5×5 conv. 64 Stride 2 lReLU |
| Reshape to 4x4x256 | 5×5 conv. 128 Stride 2 lReLU |
| 5×5 deconv. 128 Stride 2 ReLU | 5×5 conv. 256 Stride 2 lReLU |
| 5×5 deconv. 64 Stride 2 ReLU | Reshape to 4096 |
| 5×5 deconv. 3 Stride 2 Tanh | MLP 128 units Linear |
| Output image $x$ | Output latent $h$ |

Table 2: The discriminators on **SVHN**

| Global $D_{x,h,k}$ | Local $D_{x,h}$ and $D_{h,k}$ |
|---|---|
| Input $(x, h, k)$ | Input $(x, h)$ and $(h, k)$ |
| Get $x$ | Get $x$ |
| 5×5 conv. 64 Stride 2 lReLU $\alpha$ 0.2 | 5×5 conv. 64 Stride 2 lReLU $\alpha$ 0.2 |
| 5×5 conv. 128 Stride 2 lReLU $\alpha$ 0.2 | 5×5 conv. 128 Stride 2 lReLU $\alpha$ 0.2 |
| 5×5 conv. 256 Stride 2 lReLU $\alpha$ 0.2 | 5×5 conv. 256 Stride 2 lReLU $\alpha$ 0.2 |
| Reshape to 4096 | Reshape to 4096 |
| Concatenate $h$ and $k$ | Get $h$ |
| MLP 512 units lReLU $\alpha$ 0.2 | MLP 512 units lReLU $\alpha$ 0.2 |
| Concatenate features of $x$ and $(h, k)$ | Concatenate features of $x$ and $h$ |
| MLP 512 units lReLU $\alpha$ 0.2 | MLP 512 units lReLU $\alpha$ 0.2 |
| MLP 1 unit Sigmoid | MLP 1 unit Sigmoid |
|  | Concatenate $h$ and $k$ |
|  | MLP 512 units lReLU $\alpha$ 0.2 |
|  | MLP 512 units lReLU $\alpha$ 0.2 |
|  | MLP 512 units lReLU $\alpha$ 0.2 |
|  | MLP 1 unit Sigmoid |
| Output a binary unit | Output two binary units |

29 The model size and the usage of the batch normalization depend on the data. The size of $h$ in both
30 GMGAN and SSGAN is 128 and the size of $v$ in SSGAN is 8. All models are trained with the ADAM
31 optimizer with $\beta_1 = 0.5$ and $\beta_2 = 0.999$. The learning rate is fixed as 0.0002 in GMGAN and
32 0.0001 in SSGAN. In GMGAN, we use the Gumbel-Softmax trick to deal with the discrete variables
33 and the temperature is fixed as 0.1 throughout the experiments. In SSGAN, we use the same $\epsilon_t$ for all
34 $t = 1...T$ as the transformation between frames are equvariant on the Moving MNIST and 3D chairs
35 datasets. The batch size varies from 50 to 128, depending on the data.

Table 3: The generator and variant feature extractor on **3D chairs**

| Generator $G$ | Extractor $E_2$ |
|---|---|
| Input latent $(h, v_t)$ | Input frame $x_t$ |
| MLP 4096 units ReLU | 5×5 conv. 32 Stride 2 lReLU |
| Reshape to 4x4x256 | 5×5 conv. 64 Stride 2 lReLU |
| 5×5 deconv. 128 Stride 2 ReLU | 5×5 conv. 128 Stride 2 lReLU |
| 5×5 deconv. 64 Stride 2 ReLU | 5×5 conv. 256 Stride 2 lReLU |
| 5×5 deconv. 32 Stride 2 ReLU | Reshape to 4096 |
| 5×5 deconv. 3 Stride 2 Tanh | MLP 8 units Linear |
| Output frame $x_t$ | Output latent $v_t$ |

Table 4: The transition operator and invariant feature extractor on **3D chairs**

| Generator $O$ | Extractor $E_1$ |
|---|---|
| Input latent $v_1$, noise $\epsilon$ | Input video $(x_{1:T})$ |
| Concatenate $v_t$ and $\epsilon$ | Concatenate all frames along channels |
| MLP 256 units lReLU | 5×5 conv. 32 Stride 2 lReLU |
| MLP 256 units lReLU | 5×5 conv. 64 Stride 2 lReLU |
| MLP 8 units Linear | 5×5 conv. 128 Stride 2 lReLU |
| Get $v_t$ | 5×5 conv. 256 Stride 2 lReLU |
| MLP 8 units Linear | Reshape to 4096 |
| Add the features of $(v_t, \epsilon)$ and $v_t$ | MLP 128 units Linear |
| Output $(v_{1:T})$ recurrently | Output latent $h$ |

We present the detailed architectures of Graphical-GAN on the SVHN and 3D chairs datasets in the Tab. 1, Tab. 2, Tab. 3, Tab. 4 and Tab. 5, where $\alpha$ denotes the ratio of dropout. The architectures on the other datasets are quite similar and please refer to our source code.

# E   More Results of GMGAN

We present the t-SNE visualization results of GAN-G and GMGAN-L on the test set of MNIST in Fig. 1 (a) and (b) respectively. Compared with GAN-G, GMGAN-L learns representations with clearer margin and less overlapping area among classes (e.g. top middle part of Fig. 1 (a)). The visualization results support that a mixture prior helps learn a spread manifold and are consistent with the MSE results.

See Table 6 for the clustering accuracy on the CIFAR10 dataset. All the methods achieve low accuracy because the samples within each class are diverse and the background is noisy. To our best knowledge, no promising results have been shown yet in a pure unsupervised setting.

Table 5: The discriminators on **3D chairs**

| 3DCNN $D_{x_{1:T},h,v_{1:T}}$ | ConcatX $D_{x_{1:T},h,v_{1:T}}$ | Local $D_{x_t,h,v_t}$ and $D_{v_t,v_{t+1}}$ |
|---|---|---|
| Input $(x_{1:T}, h, v_{1:T})$ | Input $(x_{1:T}, h, v_{1:T})$ | Input $(x_t, h, v_t)$ and $(v_t, v_{t+1})$ |
| Get $x_{1:T}$ | Concatenate all frames along channels | Get $x_t$ |
| 4×4×4 conv. 32 Stride 2 lReLU $\alpha$ 0.2 | 5×5 conv. 32 Stride 2 lReLU $\alpha$ 0.2 | 5×5 conv. 32 Stride 2 lReLU $\alpha$ 0.2 |
| 4×4×4 conv. 64 Stride 2 lReLU $\alpha$ 0.2 | 5×5 conv. 64 Stride 2 lReLU $\alpha$ 0.2 | 5×5 conv. 64 Stride 2 lReLU $\alpha$ 0.2 |
| 4×4×4 conv. 128 Stride 2 lReLU $\alpha$ 0.2 | 5×5 conv. 128 Stride 2 lReLU $\alpha$ 0.2 | 5×5 conv. 128 Stride 2 lReLU $\alpha$ 0.2 |
| 4×4×4 conv. 256 Stride 2 lReLU $\alpha$ 0.2 | 5×5 conv. 256 Stride 2 lReLU $\alpha$ 0.2 | 5×5 conv. 256 Stride 2 lReLU $\alpha$ 0.2 |
| Reshape to 4096 | Reshape to 4096 | Reshape to 4096 |
| Concatenate $h$ and $v_{1:T}$ | Concatenate $h$ and $v_{1:T}$ | Concatenate $h$ and $v_t$ |
| MLP 512 units lReLU $\alpha$ 0.2 | MLP 512 units lReLU $\alpha$ 0.2 | MLP 512 units lReLU $\alpha$ 0.2 |
| Concatenate features of $x_{1:T}$ and $(h, v_{1:T})$ | Concatenate features of $x_{1:T}$ and $(h, v_{1:T})$ | Concatenate features of $x$ and $h$ |
| MLP 512 units lReLU $\alpha$ 0.2 | MLP 512 units lReLU $\alpha$ 0.2 | |
| MLP 1 unit Sigmoid | MLP 1 unit Sigmoid | |
| | | Concatenate $v_t$ and $v_{t+1}$ |
| | | MLP 512 units lReLU $\alpha$ 0.2 |
| | | MLP 512 units lReLU $\alpha$ 0.2 |
| | | MLP 512 units lReLU $\alpha$ 0.2 |
| | | MLP 1 unit Sigmoid |
| Output a binary unit | Output a binary unit | Output $2T - 1$ binary units |

(a) GAN-G       (b) GANGM-L

Figure 1: t-SNE visualization of the latent space on MNIST.

(a) GMGAN-L ($K = 50$) on the SVHN dataset

(b) GMGAN-L ($K = 30$) on the CIFAR10 dataset

Figure 2: Samples of the GMGAN-L on the SVHN and CIFAR10 datasets. The mixture $k$ is fixed in each column of (a) and (b).

See Fig. 2 and Fig. 3 for the complete results of GMGAN-L on the SVHN, CIFAR10 and CelebA datasets, respectively. We also present the samples of GMGAN-L with 30 clusters on the MNIST dataset in Fig. 4 (a). With larger $K$, GMGAN-L can learn intra-class clusters like "2" with loop and "2" without loop, and avoid clustering digits in different classes into the same component. GMGAN-L can also generate meaningful samples given a fixed mixture and linearly distributed latent variables, as shown in Fig. 4 (b) and (c).

# F    More Results of SSGAN

See Fig. 5 for the samples from SSGAN-L and all baseline methods. Again, all baseline methods fail to converge. See Fig. 6 for the reconstruction and motion analogy results of SSGAN-L on the Moving MNIST dataset. See Fig. 7 for the reconstruction results on the 3D chairs datasets.

Figure 3: Samples of GMGAN-L ($K = 100$) on the CelebA dataset.

(a) MNIST ($K = 30$)

(b) SVHN ($K = 50$)

(c) CelebA ($K = 100$)

Figure 4: (a): 30 mixtures of GMGAN-L on the MNIST dataset. (b) and (c): Interpolation results of GMGAN-L on the SVHN and CelebA datasets, respectively. Three endpoints are randomly sampled to construct a parallelogram and other points are linearly distributed.

Table 6: The clustering accuracy on CIFAR10 datasets.

| Algorithm | ACC on CIFAR10 |
|---|---|
| *GMM* | 20.36 ($\pm$0.69) |
| *GAN-G+GMM* | 19.19 ($\pm$0.10) |
| *GMGAN-G* | 22.63 ($\pm$1.09) |
| *GMGAN-L* (ours) | **25.14** ($\pm$2.38) |

Figure 5: Samples on the 3D chairs dataset when $T = 31$.

(a) Reconstruction          (b) Motion Analogy

Figure 6: (a) Reconstruction results. Each odd row shows a video in the testing set and the next even row shows the reconstructed video. (b) Motion analogy results. Each odd row shows a video in the testing set and the next even row shows the video generated with a fixed invariant feature $h$ and the dynamic features $v$ inferred from the corresponding test video.

Figure 7: Reconstruction results of SSGAN-L on the 3D chairs dataset.