[Reviews · NeurIPS 2018]

Reviewer 1



This paper proposes Graphical-GAN, a variant of GAN that combines the expressivity of Graphical Models (in particular, Bayesian nets) with the generative inductive bias of Generative Adversarial Networks. Using adversarial learning, the resulting hybrid generative model minimizes for a f-divergence between the joint distributions of the generative model p(X, Z) and an auxiliary encoding distribution q(X,Z). For highly structured latent variables, such as the ones considered in this work, the discriminator's task of distinguishing X,Z samples from the two distributions can be different. As a second major contribution, the work proposes a learning procedure inspired by Expectation Propogation (EP). Here, the factorization structure of the graphical model is explicitly exploited to make the task of the discriminator "easier" by comparing only subsets of variables. Finally, the authors perform experiments for controlled generation using a GAN model with a mixture of Gaussians prior, and a State-Space structure to empirically validate their approach. The paper is clearly written, easy-to-follow and achieves a good balance of intuition and formalism. The paper is closely related to prior works at the intersection of inference in GANs, in particular ALI and BiGAN. However, the key novelty of the work is the use of the formalism of graphical models for structured learning and inference. That is, instead of assuming a simple dependence of Z->X, the work looks at richer, alternate factorizations which can explicitly encode prior knowledge, such as a Gausssian Mixture Model. To the best of my knowledge, the work is technically correct. The empirical evaluation is moderately convincing (shortcomings explained in more detail later in the review). At a broader level, this work asks the following question: Does the structure of graphical models have a role to play in generative modeling? The answer is a clear yes to me, and I was highly pleased to see this work taking a step in this direction. As the authors have noted, the first work in this regime is due to Johnson et al. (done in the context of VAEs) --- however this work is the first to do for adversarially learned generative models. For the rebuttal phase, I would like to hear the author's responses to the following questions/comments: 1. Since we are using GANs, likelihood evaluation is intractable (except for invertible decoders). Then, the key benefit of Graphical-GANs over other models is controlled generation and inferring latent representations in the presence of prior information (such as the MNIST dataset can be assumed to be generated from 10 independent GANs). InfoGAN enjoys all of these properties. In L199, it is said that GMGAN doesn't need a regularization loss and is heirarchical, which makes it superior. Even though these distinctions are valid, I don't see why the Graphical-GAN approach should be preferred over InfoGAN, both in theory and experiments. In fact, Graphical-GNN requires additional information tn the form of a strcutured graphical model. On the other hand, InfoGAN seems to automatically uncover this hidden structure and hence, would be more immune to "structure mismatches" between the model and the assumed data. Can the authors comment on this and if possible, report summary experimental results comparing with InfoGAN? 2. Missing experiments -- Why are the entries in first and third column of Table 1 missing? What about clustering accuracy on CIFAR-10 for the different baselines? 3. The use of the term generalization in L275 is misleading. The only widely accepted definition of generalization I am familiar with posits that the train and test performance of a learned model should be similar to claim that the model is generalizing. This is really the opposite of generalization we see in Figure 8 and lines 275-277 where the training dataset is very different from the generated data in the length of the frames. 4. Looking at the full page of miniature samples in Figure 1 and 2 in Appendix E and Figure 5 in Appendix F tells me nothing about the performance of GMGAN-L on axis of interest. Since these are all labelled/attributed datasets, I would like to see some quantitative metrics such as average clustering accuracy, inter-intra clustering distances etc. to be able to derive a conclusion. ---- ---- Thanks for the response. I am convinced with the alternate explanation that the proposed framework can solve a diversity of tasks with much richer graphical structures, unlike InfoGAN. This reduces the significance of the MNIST experiments compared to prior work. And R2 has raised some valid concerns with the SSGAN experiments that the authors could address in a later version.

Reviewer 2



The paper introduce a idea to combine graphical models under GAN framework. The proposed idea is to match distributions locally, which is similar to conditional distribution matching, instead of learning whole joint distribution together. The generally idea is interesting to NIPS, but the evaluation is relatively weak, which may need more justification to prove the idea of the proposed algorithm. 1. In line 234, it says "GMGAN-L has less meaningless samples compared with GAN-G". This comparison is very subjective. For ambiguity I can see in Fig 2(b) is some 8 similar to 1. However, MNIST does have these distorted digits in the data. On the other hands, it seems 2(c) has more confusion on 4 and 9. The arguments above can also be challenged. The point is, it is not convincing to me to judge the performance based on Fig 2, given they also fail to correctly cluster 10 digits. For quantitative results on MNIST (table), the gap is not significant by taking std into account, so maybe MNIST is not a good example to convince people GMGAN-L is superior than GMGAN-G. 2. In "Infogan", it also learns latent code for digits. The results they reported in the paper seems correctly cluster 10 digits. I don't see any discussion about that in the draft, could you comment on that? 3. For clustering results, what's the clustering ACC on cifar10? Many columns reported in Fig4(b) and 4(c) are based on color tones, which can also be achieved by clustering based on pixel values. Then a fundamental question is, does the proposed algorithm really learn interesting clustering results? A simple two-stage baseline is we do clustering first, then train GANs. 4. For SSGAN, the baselines seem to be too weak. For example, what's the architecture of 3DCNN (I apologize if you mentioned somewhere but I missed it). If the model capacity is enough, MNIST should not be a difficult task to it. Also, instead of those proposed baselines, there are many state space models like [13], I would like to see the comparison with those. At least I expect to see the competitive performance with those VAE based models. 5. Minor comments: the figures are all too small in the paper. It is hard to check if you print it out. ==== The rebuttal addressed my concern in mixture models, so I raised my score from 4 to 5. However, there is still concern about the bad performance of 3DCNN for fair comparison. The reason mentioned in the paper is [44] use a two-stream architecture. However, in practice, although two streaming architecture brings improvement, using one stream won't results in "significantly" worse performance. For example, in [44]'s table 1. Based on human evaluation, one stream network is evaluated to be better than two-stream case in 47% testing cases. Also, the follow up work of [44] ("Generating the Future with Adversarial Transformers") is using one stream architecture. Therefore, I strongly encourage the author to investigate this issue and address this issue in the revision.

Reviewer 3



This paper proposed a flexible generative modeling framework named Graphical Generative Adversarial Networks(Graphical-GAN). The difference between this work with previous ones is they provide a general solution to utilize the underlying structural information of the data. In general, this paper presented an interesting problem setting and a corresponding solution to learn interpretable representations and generating structured samples. Here are detailed reviews: Strengths: 1. The problem addressed in this paper aims to introduce a structured recognition model to infer the posterior distribution of latent variables given observations. 2. The novelty and technical contributions are significant. The proposed learning algorithms are generally applicable to arbitrary Graphical-GAN. Weaknesses: 1. Algorithms in this paper seem very brief, which is hard to reproduce the results. 2. Since the proposed method focuses on using Gaussian mixture prior to help the model to learn a more spread latent space with less ambiguous areas, the authors may want to carefully define what is the ambiguous area, and also provide more solid numerical experiments instead of only case studies.